# Animal-Assisted Therapy as a Non-Pharmacological Approach in Alzheimer’s Disease: A Retrospective Study

**DOI:** 10.3390/ani10071142

**Published:** 2020-07-06

**Authors:** Antonio Santaniello, Susanne Garzillo, Alessia Amato, Mario Sansone, Annalisa Di Palma, Annamaria Di Maggio, Alessandro Fioretti, Lucia Francesca Menna

**Affiliations:** 1Department of Veterinary Medicine and Animal Productions, Federico II University of Naples, 80134 Naples, Italy; susannegarzillo@gmail.com (S.G.); alessiaamatovet@gmail.com (A.A.); fioretti@unina.it (A.F.); menna@unina.it (L.F.M.); 2Department of Electrical Engineering and Information Technology, Federico II University of Naples, 80125 Naples, Italy; 3Department of Fragility, Alzheimer Center “Villa Walpole”, ASL Napoli 1 Centro, 80125 Naples, Italy; dipalmaannalisa@gmail.com; 4Regional Reference Center of Urban Veterinary Hygiene (CRIUV), ASL Napoli 1 Centro, 80125 Naples, Italy; annamariadimaggio@libero.it

**Keywords:** animal-assisted therapy, reality orientation therapy, non-pharmacological therapies, patients, dog co-therapist, zootherapist veterinarian

## Abstract

**Simple Summary:**

Non-pharmacological approach represents a valid therapeutic option as an alternative or supplement to pharmacological treatments in patients with Alzheimer’s disease. A type of non-pharmacological therapy is animal-assisted therapy (AAT), where the dog is a valid support to improving the quality of life of patients. In patients with Alzheimer’s disease, interaction with animals can reduce behavioral, stress, and mood disorders, and it can also stimulate some cognitive functions and give benefits to the psychosocial sphere. The purpose of this study was to apply, over a long period of time (2012–2019), AAT interventions adapted to reality orientation therapy (ROT), in groups of patients with mild-to-moderate Alzheimer’s disease. The work aimed to stimulate neuro-cognitive functions such as spatio-temporal orientation, memory, the ability to calculate, and language and to improve the depressive state of patients through the interaction and carrying out of structured games with the dog. The results obtained in the present study show an improvement in both cognitive function and mood of patients who carried out the therapy with the dog. In conclusion, we can therefore affirm how the study conducted confirms the potential of animal-assisted therapy as a non-pharmacological therapy in the treatment of deficits deriving from Alzheimer’s disease patients.

**Abstract:**

Recently, many efforts have been made to assess the effectiveness of non-pharmacological therapies as an alternative or supportive option to conventional approaches. Specifically, animal-assisted therapy (AAT) has recently raised a great interest and large research efforts. This work represents a retrospective study carried out over seven years (from 2012 to 2019) in 127 patients with mild-to-moderate Alzheimer’s disease. The patients were divided into three groups: an experimental group that received AAT interventions adapted to the formal reality orientation therapy (ROT), a group receiving a formal ROT, and a control group that did not perform any of the previous therapies. All sessions, for all patient groups, were held weekly for a total period of six months. The evaluation of cognitive function was performed through the Mini Mental State Examination (MMSE), while the Geriatric Depression Scale (GDS) assessed the depressive state. Test administration to all patients was performed before the start of the first session (T_0_) and after the last session (T_1_). The results obtained showed an improvement in the values in the GDS and MMSE tests. The variations between the average MMSE values between T_1_ and T_0_ were 0.94 ± 0.9 (SD), 0.15 ± 0.62, and −0.42 ± 0.45 in the AAT group, ROT group, and control (CTRL) group, respectively. The variations between the average GDS values between T_1_ and T_0_ were −1.12 ± 1.17 (SD), −0.42 ± 1.21, and 0.12 ± 0.66 in the AAT group, ROT group, and CTRL group, respectively. Based on our findings, we can therefore affirm how the study carried out confirms the potential of AAT performed by Federico II Model of Healthcare Zooanthropology, and particularly its efficacy in the treatment of cognitive deficits deriving from Alzheimer’s disease.

## 1. Introduction

Alzheimer’s disease (AD) is a chronic degenerative disease that slowly and progressively destroys brain cells. This disease causes an irreversible deterioration of higher cognitive functions such as memory, reasoning, and language, to which behavioral disturbances are also added. All this leads to a total compromise of the functional state and the ability to carry out normal basic daily activities [1]. Dementia affects nearly 50 million people worldwide and the number of cases is expected to reach over 131 million by 2050 [2]. The etiological mechanisms underlying AD remain unclear but are probably influenced by environmental and genetic factors [3].

The main pathological changes observed in the brain tissue of Alzheimer’s disease are given by the increase in the levels of the amyloid-β (Aβ) peptide, which is deposited in the extracellular neuritic plaques, by the tau protein (p-tau) and by cerebral amyloid angiopathy due to the deposition of Aβ on the walls of the vessels. In addition, in this condition, there is a widespread loss of neurons and synapses [4]. Although there has been great diagnostic progress in recent decades, much progress has been made in pathogenesis and clinical practice, but the triggering factors, onset, and progression of AD remain unclear [5]. For this reason, medicine still needs more precise tools for an early diagnosis of this incurable disease [6].

Currently, pharmacological therapy is mainly based on symptomatic treatment. The approved drugs for AD improve only the symptoms of patients without changing the progression of the disease [7].

AD could not only be a brain disorder but also a systemic disease; therefore, developing a specific treatment strategy from a systemic point of view can provide a new approach to the prevention and treatment of this pathology [5].

The limited efficacy of pharmacological therapies and the plasticity of the human brain are the most important reasons for the growing interest in non-pharmacological therapies in Alzheimer’s dementia. The possibility of having alternatives to pharmacological intervention increases the number of therapeutic options, therefore non-pharmacological practices should be the first-line approach to increase these therapeutic options by offering an effective support to the pharmacological therapies.

A type of non-pharmacological therapy is animal-assisted therapy (AAT), in which animals represent an essential part to improving specific outcomes of a patient. AAT is used also to support conventional therapies, is performed by a multi-professional team with the involvement of the animal, and is customized for each patient [8].

In the geriatric population, interaction with animals not only seems to reduce behavioral disorders (e.g., agitation, aggression), stress, and mood disorders (e.g., anxiety, apathy, depression) but also stimulates some cognitive functions [9]. Previous research has shown a significant benefit in long-term memory, verbal and non-verbal communication, and sensory stimulation. Furthermore, this type of intervention satisfies some fundamental human needs such as attention and feelings of affection [10].

Menna et al. [11,12] showed how through repeated verbal, visual, and tactile multimodal stimulations, these interventions are applicable and effective in cognitive stimulation and emotional improvement of the patient, acting on mood and enabling non-medicalization of the symptom through “structured play” with the dog.

Some studies have shown how AAT can provide an alternative or an addition to pharmacological treatments to reduce the behavioral and psychological symptoms [13] as well as agitation, depression, and apathy [14] of patients with dementia. Still other studies have highlighted how this type of intervention can provide significant benefits to improve the quality of life in people with AD [15] and the psychosocial well-being in people with dementia [16]. Other studies have shown an improvement in cognitive impairment [17] and an improvement in balance [18] in patients with AD undergoing AAT with a dog.

The present work aimed to carry out a retrospective study that included all patients with mild-to-moderate AD, who in the period 2012–2019 received AAT interventions adapted to the reality orientation therapy (ROT) protocol as reported by Menna et al. [11,12].

## 2. Materials and Methods

This study was performed during several periods at two Alzheimer’s Centers in Southern Italy from January 2012 to December 2019. The head geriatrician suggested proposing the AAT interventions when patients were not inclined to perform the routine non-pharmacological therapies.

All patients gave and signed their informed consent for inclusion in the study (ISO 9001-2015 Cert. n. 317jSGQ10). In addition, the study was performed according to the Declaration of Helsinki, and the protocol was approved by the Federico II University Ethical Committee.

### 2.1. Team

According to Federico II Model of Healthcare Zooanthropology, the team was formed by the Zootherapist Veterinarian and the co-therapist dog (ISO 9001-2015 Cert. n. 317didaSGQ02) [19,20].

For the choice of dogs involved in the work of AAT, the methodology reported by Menna et al. [2,11,20] was used. The co-therapist dogs underwent regular health and behavioral checks, carried out in collaboration between our Department and Public Veterinary Service of the “ASL Napoli1”. In addition, for each session, disinfectant wipes (chlorhexidine, ethylene diamine tetra acetic acid-tromethamine (Tris-EDTA), zinc gluconate, and glycerin) were used to clean the coat, the claws, and the tail of the dog to avoid the transmission of zoonotic agents (e.g., bacteria, fungi, parasitic elements) [21,22,23,24].

### 2.2. Operative Method

#### 2.2.1. Study Design and Participants

The head geriatricians of the Alzheimer’s Centers randomly selected a total of 127 patients (95 women, 32 men) with mild-to-moderate AD and without behavioral disorders. Subsequently, all patients were divided in three groups as follows: an experimental group receiving AAT interventions adapted to the formal ROT (AAT group), a group that received a formal ROT (ROT group), and a control group that did not perform any of the previous therapies (CTRL group). The subjects had a mean age of 76.0 ± 7.1 (SD) years (range: 50–89 years) and were homogeneous for AD diagnosis and depressive status (mean ± SD of Mini Mental State Examination (MMSE): 19.7 ± 1.9 and mean ± SD of 15-item Geriatric Depression Scale (GDS): 11.9 ± 2.3, respectively).

The AAT group comprised 65 subjects (13 men, 52 women), enrolled according to the following criteria: the absence of fear or allergy the dog; the will to interact with the dog; the presence of animals in the patient’s personal history (e.g., if the patient had a dog in the past). To this group a cycle of AAT interventions based on the protocol of formal ROT [11,12] was addressed. The ROT group comprised 31 patients (9 men, 22 women), engaged exclusively in activities of formal ROT [25,26,27]. Finally, 31 patients (10 men, 21 women) formed the CTRL group participating in no stimulations.

#### 2.2.2. Methodology

All sessions (for all patient groups) were performed weekly, with a total duration of 45 min per session, for a total period of 6 months. In particular, the AAT sessions had a total duration of 45 min, of which about 20 min of activities were carried out with the co-therapist dog.

As in a session of formal ROT, the AAT adapted session aimed to stimulate cognitive functions such as attention, language skills, and spatio-temporal orientation. The activities with the dog were performed according to the ROT intervention techniques through the operative sequence shown in comparison in Table 1. In the first step, the setting structuring, presentation of the dog/zootherapist veterinarian dyad, and stimulation of cognitive functions through information and characteristics of the dog (e.g., size, coat color, hair type, eye color, and ear shape) took place; in the second step, the zootherapist worked on the patient’s orientation in space and time by structured and cadenced play activities with the dog in the setting (i.e., hide-and-seek and find the ball) and on the stimulation of memory functions through the telling a story about their own dog or other pets; in the third step, the zootherapist continued the patient’s memory stimulation (attention) through structured play with the dog per associations and understanding of language through storytelling; in the fourth step, the session was closed.

#### 2.2.3. Measurements

In the present study the neuro-cognitive functioning and the depressive state of the patients were evaluated. The evaluation of neurophysiological parameters was performed through the Mini Mental State Examination (MMSE) test widely used for the investigation of intellectual efficiency disorders and the presence of cognitive impairment [28,29]. The test is made up of 30 items (questions) that refer to seven different cognitive areas to evaluate spatio-temporal orientation, memory, attention, ability to calculate, and language. The assessment of the depressive state was carried out using the 15-item Geriatric Depression Scale (GDS), a hetero- and self-administered instrument for monitoring the degree of depression in the elderly [30]. These tests were administered to all patients enrolled at time 0 (T_0_) before the start of each intervention and at time 1 (T_1_) after the last session (six months later).

### 2.3. Data Analysis

With the aim to assess therapies’ differences, analysis of covariance change (ANCOVA-CHANGE) was used [31]. ANCOVA-CHANGE allows to manage repeated measures from several groups in order to test for differences in pre- and post-treatment changes. Computations were performed in R [32] using the package [33].

Pairwise comparison using Wilcoxon rank sum test with Bonferroni correction was used in order to assess differences between each therapy pair.

## 3. Results

A total number of 127 patients from 2012 to 2019 was included in our retrospective study. As reported in Table 2, all the enrolled patients were homogeneous for cognitive impairment and depressive symptoms, while they were heterogeneous for age and sex.

As the objective of this study was to assess the amount of variation of GDS or MMSE due to therapy, in the following using Δ (delta), we denote the variation of a measure after treatment: for example, Δ-GDS = GDS(after) − GDS(before); a positive Δ-MMSE indicates an increase of MMSE after treatment. The ANCOVA-CHANGE tested the hypothesis that, with reference to a specific measure, there were no differences among the three groups (i.e., the three therapies yield the same amount of variation). In our study, ANCOVA-CHANGE showed that there were differences between the three groups as regards Δ-GDS and Δ-MMSE with *p*-value of 1.33 × 10^−6^ and 6.56 × 10^−11^, respectively. Further, after ANCOVA it is necessary to identify which groups are different from each other (there are three possible pairwise comparisons). To do so we used the Wilcoxon rank sum test (with Bonferroni correction). First, as regards Δ-GDS, the ROT and AAT were not significantly different (*p* = 0.18), while the AAT and CTRL were different (*p* < 0.01); further ROT and CTRL were also different (*p* = 0.01). Second, as regards Δ-MMSE, all pairwise comparisons were significant (*p* = 0.01).

In Figure 1 and Figure 2, the boxplots showing Δ-GDS and Δ-MMSE are shown. First, the average Δ-MMSE (MMSE increase) in the AAT group was +0.94 ± 0.9 (SD), in the ROT group it was +0.15 ± 0.62, and in the CTRL group it was −0.42 ± 0.45. Second, the average Δ-GDS (GDS decrease) in the AAT group was −1.12 ± 1.17 (SD), in the ROT group it was −0.42 ± 1.21, and in the CTRL group it was 0.12 ± 0.66.

We can summarize the two main results as follows: first, the amount of MMSE increase due to AAT was significantly higher than ROT and CTRL; second, the amount of GDS decrease due to AAT was not statistically different from ROT although inspection of Figure 1 showed an important trend.

## 4. Discussion

The statistical evaluation of the data collected and processed with ANOVA found statistical significance; in fact, the object data demonstrate how the interaction with the dog represents a positive stimulus for the patient, measurable through an improvement of the values in the GDS and MMSE tests. The values obtained through the GDS test, administered at T_0_ and T_1_, to all the groups involved in the study, also showed an improvement in the mood tone evidenced by the decreasing variation of the values at T_1_, with a difference which is however statistically insignificant between the ROT and AAT groups. AAT, on the other hand, proved to be particularly more effective than ROT if the data relating to the MMSE test are observed. The Δ that is created between the test results detected at T_0_ and T_1_ here is particularly marked between the ROT and AAT groups.

This suggests that in the field of dementia, AAT produces surprising results compared to reality-oriented therapies that do not involve the participation of co-therapist animals in the setting.

These are tangible results, as the data measured with evaluation tests (particularly, MMSE) demonstrate how the psychomotor attitude and autonomy improve thanks to the motivation and positive feelings that the intervention of AAT can stimulate in the patient.

To our knowledge, comparative data regarding AAT with the dog adapted to ROT and aimed at Alzheimer’s patients are very limited. In the consulted literature about AAT in Alzheimer’s disease, other studies usually provide experimental designs with two groups of patients (experimental and control group) or with only the experimental group (AAT). In these studies, the cognitive and mood improvements from dog activities were measured by changes in the MMSE and GDS tests at T_0_ and T_1_. In a pilot study with the same experimental design conducted by Menna et al. [11], 50 patients were divided in three groups: AAT (adapted to ROT), ROT, and control. The intervention lasted a total of six months. The AAT group and ROT group showed a small improvement in the mood, according to GDS scores. Particularly, the AAT group improved from 11.5 at T_0_ to 9.5 at T_1_ while the ROT group improved from 11.6 at T_0_ to 10.5 at T_1_. Moreover, an improvement in cognitive impairment, as measured by the MMSE, was observed. In the AAT group, the mean value of MMSE was 20.2 at T_0_ and 21.5 at T_1_, and in the ROT group, it was 19.9 at T_0_ and 20.0 at T_1_. Differently, in the control group, the average values of both tests remained almost unmodified. A similar study performed by Menna et al. [12] highlighted the efficacy of AAT adapted to formal ROT in Alzheimer’s patients, by evaluating the GDS and MMSE values and by monitoring salivary cortisol, the latter as a measure for stress evaluation and a potential neurodegenerative disease biomarker. The intervention lasted three months. MMSE and GDS scores in the AAT group werr significantly different before and after the intervention (increased and decreased respectively); instead, the opposite occurred in the CTRL group. For the AAT group, MMSE increased and GDS decreased significantly (*p* < 0.05, test for paired data) after the whole cycle of therapy; for the CTRL group MMSE decreased while GDS increased but not significantly. In their study, Kanamori et al. [34], performed AAT for a total of six biweekly sessions. The AAT group consisted of seven subjects and the control group of 20 subjects. In a comparison between MMSE scores at baseline and those measured three months later, the average score before AAT (baseline) was 11.43 (±9.00) and 12.29 (±9.69), respectively. Differently, in the control group involving 20 patients, before the average score (baseline) was 10.20 (±7.04) and three months later it was 9.50 (±6.26). Moretti et al. [35] evaluated the effects of pet therapy on cognitive function, mood, and perceived quality of life on elderly inpatients affected by dementia, depression, and psychosis. MMSE and GDS were administered to 10 patients (pet group) and 11 controls (control group). The AAT intervention lasted six weeks. Both groups showed improvement regarding GDS and MMSE. In the pet group, depressive symptoms by GDS values decreased by 50% (from 5.9 to 2.7, *p* = 0.013), while the mean MMSE score increased by 4.5 (*p* = 0.060). Moreover, the between-group comparison showed a positive effect of pet therapy intervention on GDS (*p* = 0.070). Motomura et al. [36] conducted a study on only eight patients admitted to a local nursing home. AAT with two dogs was carried out for 1 h over four consecutive days. Although the intervention of AAT influenced the mental and apathetic state of patients, the results indicated no significant difference in the GDS and MMSE before and after therapy. Wesenberg et al. [16] performed a within-subject design with two different interventions, animal-assisted intervention (AAI) and control, and several measurement times (baseline, after three months, and after six months). Nineteen patients with dementia were included in the AAI (with a dog), and in the control intervention. Both interventions were performed as weekly group sessions, over a period of six months. Patients had moderate-to-moderately severe dementia, according to mean MMSE score at baseline of 15.18. As expected by the authors, during the six months, the MMSE mean scores decreased to a mean of 13.63. Finally, in a study by Thodberg et al. [37], a total of 100 elderly residents in four nursing homes (median age: 85.5 years) were randomly assigned to get biweekly visits for six weeks from a person accompanied by either a dog, a robot seal (PARO), or a soft toy cat. The effects of different interventions were evaluated by MMSE, Gottfries–Bråne–Steen Scale (GBS), and GDS. The visit type did not affect the MMSE (F_2.91_ = 0.35; *p* > 0.05), GBS (F_2.90_ = 0.41; *p* > 0.05), or GDS (F_2.82_ = 0.85; *p* > 0.05) recorded after the last visits. The performed tests, however, changed over the experimental period, according to the decrease in the MMSE score (S = −483; *p* < 0.05) and increase in the GBS score (t = 2.06; *p* < 0.05), both indicating an overall worsening in patients’ cognitive impairment. Interestingly, the GDS score decreased (S = −420; *p* < 0.05), showing the correlated decrease of depressive symptoms during the experimental period.

In summary, previous studies have shown results that are sometimes encouraging to use animal-assisted therapy (AAT). However, more extensive studies should be conducted to strengthen the existing evidence, by measurement of biochemical parameters such as cortisol, for example [12]. The intensity and procedures of therapy should be standardized for a better interpretation of the benefits for patients [15], considering also the main role of the dog as co-therapist [38]. In addition, it would be desirable to standardize the methodologies used in terms of Alzheimer’s diagnosis and age of the patients, always include a control group, use a wider battery of measurement tests for the cognitive impairment and mood, establish an increased frequency of sessions (preferably weekly), and plan longer-lasting interventions. In this regard, it would be desirable to evaluate the effects of the AAT in the same group of patients for periods of at least one or two years, in order to measure its possible long-term benefits.

### Limits

To interpret the data obtained from the experimental study as a whole and adequately, some intrinsic limitations of the work must be taken into consideration, such as the numerical imbalance of the patients who received the AAT, the difference between the number of male and female participants, the total number of patients analyzed, and the wide age range of patients involved. It is also necessary to consider the limit given by the duration of the study: the intervention assisted by the animals was in fact carried out in a span of only six months and, for a more detailed investigation, it would be necessary to extend this period.

## 5. Conclusions

In conclusion, the results obtained by applying the methodology proposed by the Federico II Model of Healthcare Zooanthropology, during the indicated study period, demonstrate how the patients subjected to AAT showed an improvement in both cognitive function and mood (e.g., changes in MMSE values and GDS, respectively). Therefore, we propose the use of AAT (with the dog) adapted to a protocol of formal ROT, as a non-pharmacological therapy above all in the treatment of cognitive deficits deriving from Alzheimer’s disease. In addition, this type of intervention could be particularly useful when Alzheimer’s patients show reluctance to perform daily stimulation activities or are stalled in other non-pharmacological therapies.

## Figures and Tables

**Figure 1 animals-10-01142-f001:**
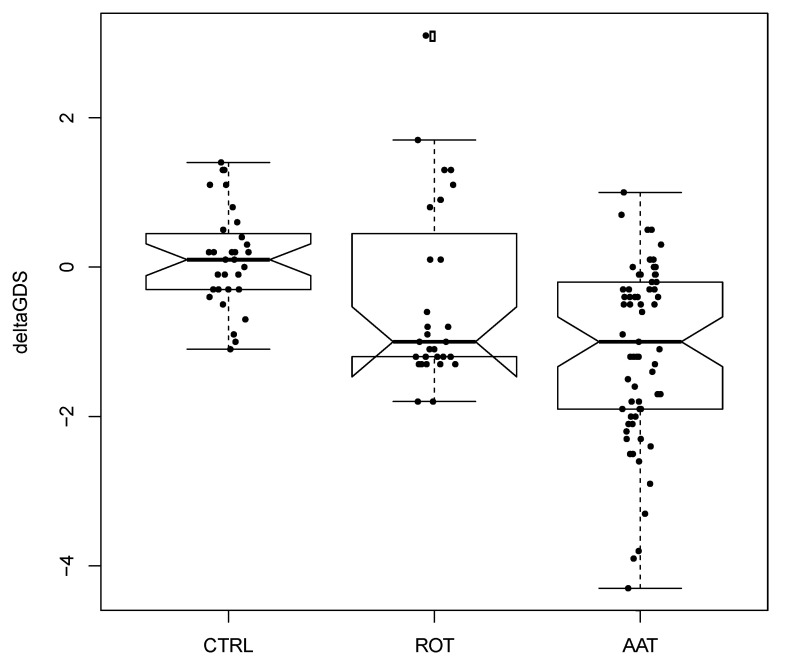
Boxplot of differences before and after treatment for Geriatric Depression Scale (GDS). Each black dot is a patient. Empty circles are outliers with respect to the main distribution. Horizontal solid line represents the median of distribution. Abbreviations: CTRL = control; ROT = reality orientation therapy; AAT = animal-assisted therapy.

**Figure 2 animals-10-01142-f002:**
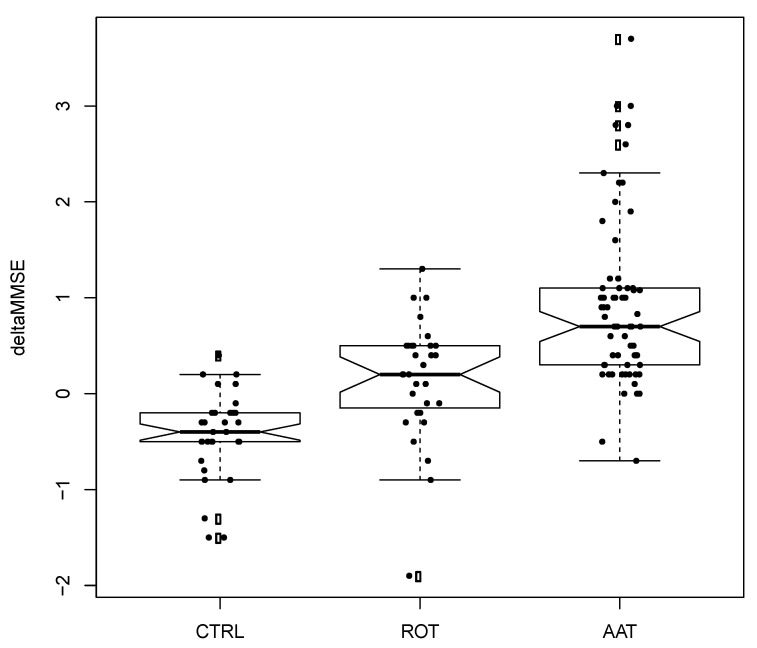
Boxplot of differences before and after treatment for Mini Mental State Examination (MMSE). Each black dot is a patient. Empty circles are outliers with respect to the main distribution. Horizontal solid line represents the median of distribution. Abbreviations: CTRL = control; ROT = reality orientation therapy; AAT = animal-assisted therapy.

**Table 1 animals-10-01142-t001:** Operational sequence and duration of each animal-assisted therapy (AAT) session with the dog addressed to the patients based on the formal reality orientation therapy (ROT) protocol by Menna et al. [11].

	Formal ROT	AAT Session	Duration
**Step 1**	1. Structuring the setting 2. Presenting the therapist/patient 3. Stimulating cognitive function	1. Structuring the setting 2. Presenting the zootherapist/dog/patient 3. Stimulating cognitive function through repeated requests for information about the dog (e.g., name, breed, age, and sex)	10′
**Step 2**	1. Temporal orientation (day, month, year, and season) 2. Spatial orientation (place, structure, floor, room, city, country, and region) 3. Stimulation of memory	1. Temporal orientation (day, month, year, and season) 2. Spatial orientation (place, structure, floor, room, city, country, and region); 5 min of play structured activities with the dog (e.g., hide the ball) 3. Stimulation of memory by telling a story about their own pets	15′
**Step 3**	1. Stimulation of memory (attention) 2. Understanding of language (story)	1. Structured play/interaction with the dog (attention: fetch, hide the ball, caring for the dog) 2. Understanding of language (story: giving the dog commands and waiting for the execution of the command)	15′
**Step 4**	1. Closing speech (ritualized)	1. Closing speech (ritualized: washing hands)	5′

**Table 2 animals-10-01142-t002:** Summary of the characteristics of the study population.

Group	No. of Patients		MMSE ^1^ (Mean ± SD)	GDS ^2^ (Mean ± SD)	Age (Years) Mean ± SD ^9^
	F ^6^	M ^7^	T_0_ ^8^		
AAT ^3^	52	13	19.2 ± 2.39	11.8 ± 2.95	76.6 ± 5.0
ROT ^4^	22	9	20.1 ± 0.94	12.4 ± 1.46	75.6 ± 6.0
CTRL ^5^	21	10	20.4 ± 0.53	11.5 ± 0.73	75.0 ± 6.3
Total	95	32	19.7 ± 1.9	11.9 ± 2.3	76 ± 5.8

^1^ MMSE = Mini Mental State Examination, ^2^ GDS = Geriatric Depression Scale; ^3^ AAT = animal-assisted therapy; ^4^ ROT = reality orientation therapy; ^5^ CTRL = control; ^6^ F = Females; ^7^ M: Males; ^8^ T_0_: Time 0; ^9^ SD: Standard Deviation.

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
