# Peer review of "Animal-Assisted Therapy as a Non-Pharmacological Approach in Alzheimer’s Disease: A Retrospective Study"

_animals, 2020, doi:10.3390/ani10071142_

Round 1

Reviewer 1 Report

Many thanks for your interesting paper I enjoyed reading it although it has some minor limitations it is of interest internationally to readers. Here are some minor changes.

Line 119 typo

Line 121 you need to replace fingertips with claws if you are referring to the canine participant!

Line 130 your sample is heterogeneous for age not homogeneous as this is a wide age range which could be mentioned in your limitations section.

Line 172 replace was with were.

You need paragraphs in your discussion to help readers absorb the volume of information being covered.

Reviewer 2 Report

The works directed to the treatment of Alzheimer's disease are always very interesting and necessary.

I think this retrospective study would be stronger if the treatment period was extended to one or two years.

I consider that the AAT sessions need to be clarified

I think it necessary that the results should be clarified. In their current version they are difficult to interpret

The discussion should be clearer and easier to understand

I believe that the conclusion of the study is written in a confusing way and mixed with data from other studies.
I think it should be modified

Round 2

Reviewer 2 Report

The questions raised have been satisfactorily answered   My suggestions have been included in the new manuscript   The manuscript has improved in clarity and quality. Now it is more understandable   I like how the conclusion is now   Thanks